# OpenReview forum: "MixAT: Combining Continuous and Discrete Adversarial Training for LLMs"
_NeurIPS.cc/2025/Conference — NeurIPS 2025 poster_

### Official Review · Reviewer_KJJ1 · 2025-06-15

**Clarity:** 4
**Significance:** 4
**Originality:** 4
**Rating:** 6
**Confidence:** 5

**Summary:**

Some past works do LLM adversarial training using discrete adversarial inputs. These are less efficient but more on-distribution for many types of real-world attacks. Others use continuous adversarial inputs. These are more efficient, but more off distribution for many real-world attacks. This paper asks if we just did discrete and then continuous attacks one after another for adversarial training. They introduce this method, analyze cosine similarity by the embeddings produced by their attacks, and test their method on a bunch of models and evals. They also do ablation tests and some other useful experiments on quantization, transferability, etc.

**Questions:**

None in particular. I think I have a good understanding of this type of paper, and I think this particular paper is very clear.

**Ethical Concerns:**

["NO or VERY MINOR ethics concerns only"]

**Final Justification:**

I think this is a great paper (see below).

**Limitations:**

MixAT is really more of a family of methods than a single method. It looks like this paper has good success with PAP + CAT, but there are a lot more possibilities.

**Quality:**

4

**Strengths And Weaknesses:**

S1: I think it's a great idea for a paper. I hope I don't see any other reviewers complain and make assertions about a lack of novelty. I work closely in the space, and I am very familiar with the methods and goals of the paper, and I think this type of work is nontrivial, non-obvious, good, and needed.

S2: Not a substantive thing -- but just a comment on design. I really like figure 1 and the fuzzy text.

S3: Pretty good diversity across experiments in model, size, evals, and baselines. I can tell this was a ton of work by the authors.

S4: Well written

S5: I like the analysis of costs in table 2.

S6: I think this paper overall does very well what it sets out to do -- making a strong case that MixAT approaches belong in the SOTA toolbox for making more robust LLMs. I think the paper is of really clear value, and I would be willing to go to bad for this paper.

W1: There is a claim in the abstract that, to my knowledge, is generally believed to be false: "As these relaxations do not correspond to discrete input tokens, such latent training methods often leave models vulnerable to a diverse set of discrete attacks." All three of the papers cited in the "Continuous Adversarial Training" paragraph of related works argue all give evidence against this claim. It's good and fine to hypothesize that discrete and continuous adversarial training, complement each other, and it's also good and fine to hypothesize that discreet adversarial training on a certain set of attacks is likely to improve generalization to similar input-space attacks, but I know of no evidence to suggest that training on latent/embedding attacks results in poor generalization due to the attacks not corresponding to actual tokens.

W2: In section 2.5, I recommend citing recent work from anthropic on constitutional classifiers.

W3: In fig 6, please extend the caption to explain the acronyms and fig. Captions should be stand-alone explanations, not just labels. See also figure 4 from [Che et al. (2025)](https://arxiv.org/abs/2502.05209) who do different but closely related analysis of the similarity of different attacks. It could also be useful related work to discuss in section 2.2.

W4: This might be a major pain in the ass, but consider replacing table 1 with heat-colored text or a heatmap. As is, I have to stare at it too long to get the jist. And I buy the topline result that, overall on average, Mixat is good (I hope no other reviewers complain about how it doesn't pareto dominate), but it would be much easier from a heat-coded version.

W5: Consider also adding two more columns to table 1 giving the mean delta for utility score over baseline and the mean delta for ASR below baseline. If you don't color-code the entire table, you could at least colormap the values in these new columns.

---

> ### Author Rebuttal · Authors · 2025-07-31
>
> We thank Reviewer $\textcolor{purple}{KJJ1}$ for their thorough review and positive assessment of our work! We are happy to hear that the reviewer finds our work and MixAT to be non-trivial, relevant, and an essential part of the LLM robustness toolbox. Below, we individually address the reviewer’s questions and comments.
>
>
> **Q16: The Abstract claim about the vulnerability of continuous attack training methods to discrete attacks seems overstated. Could the authors comment on this?**
>
> We thank the reviewer for pointing this out and agree that our original wording may have overstated the relationship between continuous relaxations and discrete attack vulnerabilities. Our intention was not to claim a direct causal link, but rather to highlight an empirical observation: in our experiments, models trained with continuous adversarial perturbations (e.g., in the embedding space) may still be susceptible to a diverse set of discrete attacks, even if individual attack success rates are low.
>
> This was actually one of the initial motivations behind the introduction of the ALO-ASR metric, which captures the worst-case failure across a suite of attacks. We believe this complements existing robustness metrics and underscores the importance of evaluating model defences against diverse, realistic threats, even when the models show robustness against individual attacks.
>
> We will revise the sentence in the abstract to better reflect this nuance and avoid implying a general claim, as follows:
>
> Despite their effectiveness and generalization capabilities, training with continuous perturbations does not always capture the full spectrum of vulnerabilities exploited by discrete attacks.
>
> We are happy to further discuss this with the reviewer in case they have particular ideas on how to best describe our observed phenomenon.
>
>
> **Q17: Would the authors consider discussing recent work from anthropic on constitutional classifiers, and recent work from Che et al. in the Related Work Section?**
>
> Certainly. We thank the reviewer for suggesting this and are happy to include an additional paragraph on model-augmenting (Guards, Classifiers, …) defenses against LLM jailbreaks as they become an increasingly important part of practical model deployments. Further, we are currently looking into incorporating the experiment from Che et al. in our own work.
>
>
> **Q18: Could the authors extend the caption of Fig 6 from App. to explain the acronyms?**
>
> Thank you for pointing this out! We will revise the caption to explain all acronyms in the updated version of the work. We will also make sure that the remaining figure captions contain full descriptions of the associated experiments.
>
>
> **Q19: Would the authors consider replacing Table 1 with heat-colored text or a heatmap?**
>
> Sure. We thank the reviewer for this great suggestion. We have considered various variants for the main table, including splitting, but for the initial version, we settled for this variant. Color-coding the values in the table might be a great way to increase readability, and we are currently experimenting with various ways to incorporate this in the next revision of the work.
>
>
>
> **Q20: Would the authors consider adding two more columns to Table 1 giving the mean delta for utility score over baseline and the mean delta for ASR below baseline?**
>
> We considered including an average Utility column in Table 1, but decided against it because the utility metrics vary widely in scale and sensitivity across different tasks and models. Averaging them could obscure important differences and lead to misleading conclusions (since the XSTest scores show the highest variance, the average would mostly follow the XSTest trends). Instead, we decided to first and foremost report individual scores to allow for a more accurate and nuanced comparison across attacks (allowing readers to select the metrics important for them - e.g., multiple choice answering vs text generation performance vs safety-specific utility benchmarks). However, we can consider changing utility scores with deltas from the baseline as this might look more informative, though the issue with averaging the deltas would be the same (XSTest would still have the highest variance).
>
> On the other hand, for a unified robustness metric we have proposed the At Least One ASR (ALO-ASR) instead of Average ASR. ALO-ASR provides a more realistic view than just averaging the scores obtained by different attacks, for two reasons: 1. ALO-ASR better reflects the real threat model where a malicious actor can try multiple attack techniques for each sample of text, until obtaining at least one harmful output. 2. ALO-ASR provides a more realistic weighting (we no longer over-emphasize easier samples where methods constantly succeed) across different harmful requests with varying hardness, resulting in a clearer model comparison.
>
> We hope that adding color-coded meaning to the tables will help visualise both the individual results and the overall (averaged) trends. However, if the reviewer considers this would improve the understanding of our claims and results, we are happy to include the requested Average Utility and Average ASR scores in the updated manuscript.

---

> > ### Comment · Reviewer_KJJ1 · 2025-08-02
> > **Reply**
> >
> > Thanks. I don't have much to litigate. I suppose I still see nothing wrong with adding two more columns to table 1 like I said, but this isn't a big deal. Say less. I hope this paper is accepted. I'll hold at at 6.

---

### Official Review · Reviewer_S3HQ · 2025-06-20

**Clarity:** 3
**Significance:** 2
**Originality:** 2
**Rating:** 4
**Confidence:** 4

**Summary:**

The paper proposes MIXAT, a hybrid adversarial training strategy for large language models that combines discrete prompt attacks (e.g., PAP paraphrasing) and continuous embedding space perturbations. The authors introduce the At Least One Attack Success Rate (ALO-ASR) metric to quantify worst-case vulnerability and report that MIXAT achieves substantially lower ALO-ASR (< 20%) than prior methods (e.g., CAT, R2D2) while maintaining comparable utility across benchmarks (e.g., ARC, MMLU)

**Questions:**

Table 1 reports a long list of individual utility and ASR results across multiple tasks and attacks, but lacks an average performance column (e.g., mean utility or mean ASR).

**Ethical Concerns:**

["NO or VERY MINOR ethics concerns only"]

**Final Justification:**

Thanks for the authors' detailed response. Most of my concerns are well solved. I thereby raise my rate to 4.

**Limitations:**

See the weaknesses.

**Paper Formatting Concerns:**

No formatting concerns.

**Quality:**

2

**Strengths And Weaknesses:**

Strengths:

1. The motivation and problem formulation of this paper are quite clear. The paper identifies key limitations of purely discrete versus purely continuous adversarial training and formulates a unified perturbation set in Eq. (8).

2. This paper provides a comprehensive evaluation across a diverse suite of attacks (PAP, TAP, GCG, AutoDAN, etc.) and multiple LLM backbones, using both standard benchmarks and the proposed ALO-ASR metric.

Weakness:

1. MIXAT simply merges two well-known ideas—discrete adversarial training (e.g., PAP-AT [2]) and continuous embedding perturbations (e.g., CAT [7])—without introducing new attack or defense techniques. The main contribution is their combination, which may limit its novelty to current LLM robustness research.

2. In Section 4.3, the authors report a “MIXAT + GCG” variant that mixes GCG attacks during training to further reduce ALO-ASR. However, this raises fairness concerns: most baselines are not allowed to train on GCG, so the comparison becomes uneven.

3. Different attack bounds. The author only reported the performance under certain attack hyperparameters. Tighter or looser attack bounds are suggested to enhance the experiments.

---

> ### Author Rebuttal · Authors · 2025-07-31
>
> We thank Reviewer $\textcolor{blue}{S3HQ}$ for their review and suggestions. We are happy to hear Reviewer $\textcolor{blue}{S3HQ}$ finds our problem well motivated and our corresponding evaluation comprehensive. Below, we individually address the reviewer’s outstanding comments.
>
> **Q12: Is MixAT technically novel? It seems that it is only a combination of existing methods.**
>
> Yes, we believe MixAT is technically novel. While our method builds upon existing works and ideas, we want to emphasize that MixAT does not represent a trivial combination of continuous and discrete adversarial training (which we evaluate as a separate baseline (DualAT) in our work), a point also raised by Reviewer $\textcolor{purple}{KJJ1}$. Instead, MixAT proposes joint discrete and continuous attacks (See Figure 1 in the paper), significantly widening the attack domain and providing a natural extension over approaches that so far have only been used separately in adversarial LLM training. Notably, while prior work in adversarial training in other domains, such as image classification, has shown that combining different attack strategies during training improves the generalization of the defense capabilities of the network, MixAT is the first work to propose this for adversarial LLM training, achieving state-of-the-art results.
>
>
> **Q13: Could you elaborate on why you compared "MIXAT + GCG" to other baselines that weren’t trained with GCG attacks?**
>
> Sure. We want to clarify that MixAT + GCG is not our main result but rather an ablation study intended to explore the impact of training with a more diverse set of attacks. Importantly, R2D2 is also trained with GCG samples, so including MixAT + GCG in the comparison helps contextualize performance across methods that leverage similar data. Our end goal is to build models that generalize to a wide range of attacks, and this ablation highlights that increasing attack diversity during training can be an effective step toward that goal.
>
>
> **Q14: Could the authors provide some experiments for varying the hyper-parameters of the continuous PGD attack?**
>
> Yes. Below, we show how scaling the $\epsilon$ (0.5x, 2x), the number of steps (0.5x, 2x), and the step-size (0.5x,2.0x) of the continuous PGD attacks used in MixAT affects the robustness and performance of the trained models. Note that for a fair comparison we scaled the step-size along with the perturbation budget $\epsilon$ and we also scaled the step-size inversely with respect to the number of steps.
>
> | Zephyr Model Utility | ARC-E | ARC-C | MMLU | Harmless | MT-Bench | XSTest |
> |---|---|---|---|---|---|---|
> | Mix-AT Original | 81.4 | 54.0 | 55.8 | 97.5 | 54.3 | 74.0 |
> | MixAT 0.5x epsilon 0.5x lr | 82.0 | 55.2 | 56.4 | 97.5 | 53.1 | 63.2 |
> | MixAT 2x epsilon 2x lr | 81.9 | 53.8 | 55.5 | 92.5 | 51.2 | 81.6 |
> | MixAT 2x steps 0.5x lr | 81.8 | 55.1 | 56.0 | 95.0 | 54.5 | 69.6 |
> | MixAT 0.5x steps 2x lr | 81.4 | 54.5 | 56.1 | 97.5 | 52.6 | 56.8 |
>
> | Zephyr Model ASR | Direct R | PAP | TAP | PAIR | AutoDAN | GCG | H Jail | ALO |
> |---|---|---|---|---|---|---|---|---|
> | Mix-AT Original | 0.0 | 0.0 | 0.0 | 0.0 | 0.0 | 12.5 | 5.0 | 15.0 |
> | MixAT 0.5x epsilon 0.5x lr | 2.5 | 0.0 | 2.5 | 2.5 | 0.0 | 12.5 | 2.5 | 12.5 |
> | MixAT 2x epsilon 2x lr | 0.0 | 0.0 | 5.0 | 5.0 | 0.0 | 2.5 | 0.0 | 12.5 |
> | MixAT 2x steps 0.5x lr | 0.0 | 7.5 | 7.5 | 20.0 | 0.0 | 15.0 | 0.0 | 32.5 |
> | MixAT 0.5x steps 2x lr | 0.0 | 2.5 | 10.0 | 10.0 | 0.0 | 10.0 | 2.5 | 20.0 |
>
> Overall, we observe consistent results on the utility benchmarks with only minor and to be expected deviations from the original values. For robustness, this trend continues, however, we find that generally the chosen tradeoff of original MixAT performs on the top end of all tested configurations, with especially high-step low learning rate experiencing a stark drop in ALO robustness. We plan to conduct further hyperparameter experiments here to analyze the trends better, and we will include them in the next revision of the paper.
>
> **Q15: Could the authors include an average Utility or average ASR column in Table 1?**
>
> We considered including an average Utility column in Table 1, but decided against it because the utility metrics vary widely in scale and sensitivity across different tasks and models. Averaging them could obscure important differences and lead to misleading conclusions (since the XSTest scores show the highest variance, the average would mostly follow the XSTest trends - e.g., multiple choice answering vs text generation performance vs safety-specific utility benchmarks). Instead, we only report individual scores to allow for a more accurate and nuanced comparison across attacks.
>
> On the other hand, for a unified robustness metric our work proposes the At Least One ASR (ALO-ASR) instead of Average ASR. ALO-ASR provides a more realistic view than just averaging the scores obtained by different attacks, for two reasons: 1. ALO-ASR better reflects the real threat model where a malicious actor can try multiple attack techniques for each sample of text, until obtaining at least one harmful output. 2. ALO-ASR provides a more realistic weighting (we no longer over-emphasize easier samples where methods constantly succeed) across different harmful requests with varying hardness, resulting in a clearer model comparison.
>
> If the reviewer believes that average Utility and average ASR would improve the understanding of our claims and results, we are happy to include the requested scores in the updated manuscript.

---

### Official Review · Reviewer_WP5s · 2025-07-03

**Clarity:** 3
**Significance:** 3
**Originality:** 2
**Rating:** 5
**Confidence:** 4

**Summary:**

This paper proposes an improvement to adversarial training for LLMs. Existing adversarial training works either optimize the adversarial perturbation in either the token embedding space, the latent space of the model, or the discrete tokens that are input to the model. This results in a lack of robustness to a particular type of attack for each of these methods. Consequently, this work, MixAT, suggests combining both discrete and embedding-space attacks during adversarial training. The authors provide convincing evidence that MixAT achieves both higher robustness and higher utility than the baseline defenses.

**Questions:**

1. There is a comparison between MixAT and static MixAT where the attacks used during training are supposed to be pre-computed and not generated on the fly. In my understanding, PAP can already be computed and is not dependent on the target model. So the only part that becomes static is the continuous AT (i.e., adversarial embeddings are computed once on the base model alone)? I’m quite surprised this even works at all. Does this work because you do LoRA and so the embedding table does not change?
2. Continuing from the first question, I wonder what happens if you also freeze the GCG suffixes, generating them only on the base model, and use it to train MixAT + static GCG. MixAT + dynamic GCG seems to be very good but suffers from high computation cost. I wonder if using static GCG finds a good trade-off.
3. How do you evaluate a response as harmful or not? Do you use one of the classifier that comes with HarmBench?
4. L315 mentions that “the quantization slightly improves the robustness.” Do you have any explanation for this?

**Ethical Concerns:**

["NO or VERY MINOR ethics concerns only"]

**Final Justification:**

I have no new concerns after the rebuttal, and my original questions are sufficiently addressed.

**Limitations:**

yes

**Quality:**

3

**Strengths And Weaknesses:**

## Strengths

1. The knowledge around adversarial training for LLMs has been sparse despite its importance in practice. I believe that this study contributes substantially to this body of knowledge, showing benefits in combining more than one types of attacks.
2. The experiments are comprehensive and convincingly show the improvement from MixAT across different base models and hyperparameters. I appreciate the ablation studies and the detailed reports on multiple utility benchmarks, multiple attacks, effects of randomness, as well as the runtime/computation cost. I also like the comparison between static and dynamic AT.
3. Overall, the presentation is good; the paper is well-written. There are lots of numbers to parse in Table 1 and in the paper so I would also appreciate some simplifications, but overall, I have no trouble understanding and learning from the paper.

## Weaknesses

The paper has no glaring weakness in my opinion, but I do have several questions in the next section.

1. The biggest point I have to say is in the novelty of the method. MixAT essentially enhances a well-known adversarial training algorithm by simply combining two different types of attacks against LLMs. That said, the paper has made up for this weakness very well by figuring out many tiny technical pieces that make this defense work (and also report the ablations) and by showing thorough empirical results.
2. Another small point is that MixAT does not perform well on XSTest but is comparable to the baselines. When measuring utility, I’d put more weights on XSTest than generic benchmarks that do not test safety capabilities of the model.

---

> ### Author Rebuttal · Authors · 2025-07-31
>
> We thank Reviewer $\textcolor{green}{WP5s}$ for the thorough review. We are happy to hear Reviewer $\textcolor{green}{WP5s}$ finds our work contributes substantially to the body of knowledge around adversarial training of LLMs, and that they find our experimental evaluation convincing and comprehensive. Below, we address the reviewer’s outstanding questions.
>
> **Q7: In the Static MixAT experiment, is the Continuous attack part the only one that is made static? Could the authors provide some intuition on why training with static continuous attacks work? Is the token embedding layer frozen during fine-tuning with LoRA adapters?**
>
> Yes, indeed in the Static MixAT experiment, the PAP-generated attacks are already model-agnostic, so it only makes sense to consider the Continuous part of the attacks as being static. Our intuition is that even when training with static continuous adversarial examples generated for the base model, these samples still have an adversarial value for the model that is being trained, mostly because our model is being fine-tuned from a pretrained model and therefore it is not substantially different from the base model. Informally, at least for the first few training steps, the difference between the model trained with statically vs dynamically generated adversarial examples should not be meaningful. As the training progresses, the two models should diverge and the statically generated attacks should get less potent, as the model starts overfitting them. However, our fine-tuning procedure is very short compared to the full pre-training of the base model, and therefore, we hypothesise that the model does not have enough time to drift significantly from the dynamically trained model. This is in contrast with most works in adversarial training for Computer Vision applications, where the models are adversarially trained from a random initialization for which training with static attacks would be meaningless.
>
> Finally, the token embedding layer is indeed frozen during LoRA training, keeping the continuous embedding attacks aligned with the model. We note that, it is also possible to freeze the token embedding layer even when fine-tuning without LoRA.
>
>
> **Q8: Would training MixAT + GCG on statically generated GCG suffixes still show improvements? Would this reduce the computational time?**
>
> Thank you for the interesting question! In theory, one would expect that training MixAT + GCG on statically generated GCG suffixes should still show improvements. Prompted by the reviewer's suggestion, we performed this experiment and show the results in the tables below.
>
> | Zephyr Model | ARC-E | ARC-C | MMLU | Harmless | MT-Bench | XSTest |
> |---|---|---|---|---|---|---|
> | CAT | 78.2 | 51.1 | 54.8 | 97.5 | 55.4 | 50.8 |
> | original MixAT (PAP + CAT) | 81.4 | 54.0 | 55.8 | 97.5 | 54.3 | 74.0 |
> | MixAT + GCG | 81.6 | 54.5 | 55.9 | 92.5 | 54.6 | 56.4 |
> | MixAT + static GCG | 82.1 | 55.2 | 56.1 | 97.5 | 53.4 | 60.0 |
>
> | Zephyr Model ASR | Direct R | PAP | TAP | PAIR | AutoDAN | GCG | H Jail | ALO |
> |---|---|---|---|---|---|---|---|---|
> | CAT | 2.5 | 40.0 | 42.5 | 42.5 | 2.5 | 5.0 | 5.0 | 70.0 |
> | original MixAT (PAP + CAT) | 0.0 | 0.0 | 0.0 | 0.0 | 0.0 | 12.5 | 5.0 | 15.0 |
> | MixAT + GCG | 2.5 | 0.0 | 2.5 | 5.0 | 0.0 | 2.5 | 2.5 | 7.5 |
> | MixAT + static GCG | 2.5 | 0.0 | 2.5 | 12.5 | 2.5 | 5.0 | 2.5 | 15.0 |
>
> | Llama3 Model Utility | ARC-E | ARC-C | MMLU | Harmless | MT-Bench | XSTest |
> |---|---|---|---|---|---|---|
> | CAT  | 79.7 | 50.9 | 58.0 | 65.0 | 65.7 | 48.4 |
> | original MixAT (PAP + CAT) | 80.4 | 50.1 | 59.1 | 85.0 | 55.6 | 40.0 |
> | MixAT + GCG | 80.1 | 48.7 | 58.5 | 92.5 | 55.2 | 47.6 |
> | MixAT + static GCG | 79.1 | 48.4 | 56.9 | 87.5 | 54.2 | 40.0 |
>
> | Llama3 Model ASR | Direct R | PAP | TAP | PAIR | AutoDAN | GCG | H Jail | ALO |
> |---|---|---|---|---|---|---|---|---|
> | CAT | 0.0 | 30.0 | 47.5 | 70.0 | 0.0 | 7.5 | 5.0 | 82.5 |
> | original MixAT (PAP + CAT) | 0.0 | 0.0 | 2.5 | 2.5 | 0.0 | 22.5 | 0.0 | 25.0 |
> | MixAT + GCG | 0.0 | 0.0 | 5.0 | 7.5 | 0.0 | 2.5 | 0.0 | 15.0 |
> | MixAT + static GCG | 0.0 | 0.0 | 0.0 | 2.5| 0.0 | 2.5 | 0.0 | 5.0|
>
> Our results show that incorporating static GCG samples into MixAT training indeed shows similar benefits with the dynamically generated ones.
>
> However, note that statically generating all the GCG samples for the whole training set takes 33.5 hours on H200 GPU for Llama 3 8B and 36.6 hours for Zephyr, which makes this singular experiment significantly more expensive than our dynamic MixAT + GCG training. On the other hand, once we have generated the static GCG samples, we can run multiple training runs incorporating them. See, for example, **Q1** of Reviewer $\textcolor{orange}{zGMj}$, where we trained a variant of MixAT that only uses GCG samples and Continuous Attacks.
>
> **Q9: How do you evaluate a response as harmful or not? Do you use one of the classifiers that comes with HarmBench?**
>
> Yes. Our evaluation uses the cais/HarmBench-Llama-2-13b-cls model released alongside HarmBench on HuggingFace to determine which responses are toxic. We will clarify this in the updated manuscript.
>
>
> **Q10: Could the authors provide any hypotheses for why model quantization slightly increases robustness?**
>
> Yes. We think that model quantization creates sharper ‘decision boundaries’ between refusal and harmful responses. This might cause slightly harder optimization problems when searching for adversarial examples. On the other hand, quantized models also have slightly lower utility scores, which might impact the model’s ability to *correctly* output harmful responses. We find this question very interesting and inspiring, but a thorough analysis on the impact of quantization on model robustness is outside the scope of our work.
>
> **Q11: Should we put more weight on XSTest than generic benchmarks that do not test the safety capabilities of the model?**
>
> We think this is a highly relevant question. Correctly evaluating LLMs in general, and robust LLMs in particular, is still a very open research problem. We agree that XSTest provides a very meaningful signal for assessing a model’s ability to distinguish between real and fake harmful requests, and we found it very useful in our evaluations. However, XSTest is essentially a binary classification task, where any non-refusal answer is scored, regardless of its quality, which may not capture the full complexity of model behavior. We believe it’s also important to ensure that the model retains its ability to generate correct and relevant answers, which is why we also consider more established benchmarks in general LLM evaluation. Ultimately, we hope that our quite extensive evaluation can provide important pointers on how LLM robustness evaluations should develop.

---

### Official Review · Reviewer_zGMj · 2025-07-05

**Clarity:** 3
**Significance:** 3
**Originality:** 3
**Rating:** 4
**Confidence:** 4

**Summary:**

This paper introduces MIXAT, a novel adversarial training (AT) method designed to enhance the robustness of Large Language Models (LLMs) against jailbreaking attacks. The authors identify a critical trade-off in existing AT methods: discrete, token-level attacks (e.g., GCG, PAP) are effective but computationally prohibitive, while continuous, embedding-space attacks (e.g., CAT) are efficient but fail to generalize to realistic, discrete threats.

MIXAT elegantly bridges this gap by proposing a two-stage adversarial sample generation process. First, it applies a low-cost discrete attack (specifically, PAP-based rephrasing) to a harmful prompt to generate a "seed" adversarial example. Second, it applies a continuous, embedding-space perturbation on top of this discrete seed. This combined approach aims to explore the adversarial space more effectively and efficiently.

The authors conduct a comprehensive evaluation on several state-of-the-art LLMs (Zephyr, Llama3, Qwen) and a wide range of attacks. They introduce a new metric, "At Least One Attack Success Rate" (ALO-ASR), to measure worst-case vulnerability. The results compellingly show that MIXAT achieves a superior robustness-utility trade-off, significantly reducing ALO-ASR to <20% compared to baselines (>50%), while maintaining high utility and incurring minimal computational overhead compared to discrete methods. The paper further provides insightful analyses on the effects of quantization, LoRA scaling, and temperature on model robustness.

**Questions:**

1. Could you elaborate on the intuition for why the combined perturbation space (R(X) + δ) is so much more effective than the union of the two spaces used in DUALAT? The paper states it "better covers the adversarial embedding space," but a deeper explanation of the underlying geometry or optimization landscape would be insightful.
2. Regarding the inclusion of GCG samples in MIXAT + GCG, the training cost increased 5x. Was this increase primarily due to the GCG sample generation, or did the training on these combined embeddings also converge more slowly?
3. In Figure 4, the utility of MIXAT-trained models is consistently higher than or equal to CAT-trained models across different LoRA scaling factors λ, even though both are being fine-tuned. Do you have a hypothesis for why MIXAT's training regimen is less detrimental to model utility compared to CAT?

**Ethical Concerns:**

["NO or VERY MINOR ethics concerns only"]

**Final Justification:**

I am not very familiar with the field of LLM safety, so I took some time to get up to speed on the relevant background while reading this paper. Based on my understanding, this work demonstrates a quality suitable for NeurIPS. However, I am probably not as excited by this work as a reviewer who would give it a score of 6. Therefore, I feel this work may not merit such a high rating; in my opinion, a score of 6 should be reserved for work that benefits the entire community. The authors' rebuttal was helpful in clarifying some of my earlier confusion, which is great. Overall, however, I believe my score is largely appropriate. I will keep my score unchanged and increase my confidence in this positive rating.

**Limitations:**

yes

**Quality:**

4

**Strengths And Weaknesses:**

## Strengths
1. The core idea of MIXAT—centering continuous perturbations around discrete adversarial seeds (R(X) + δ)—is both simple and remarkably effective. It provides a principled way to combine the strengths of both discrete and continuous AT. The comparison against the simpler DUALAT method (R(X) ∪ B(x, ε)) successfully demonstrates that the specific formulation of MIXAT is crucial to its success.
2. The experimental validation is a major strength of this paper and sets a high standard for future work in this area.

## Weaknesses

1.  The paper primarily uses PAP for generating the discrete seeds, citing its low cost and effectiveness. However, the performance of MIXAT could be dependent on the quality and diversity of these seeds. It would be beneficial to include a brief discussion or a small-scale experiment on how the choice of the discrete attack method (e.g., using a few steps of a different, non-rephrasing attack as a seed) impacts the final robustness. This would help clarify whether MIXAT's success is tied specifically to PAP or to the general principle of discrete seeding.
2. The authors perform extensive ablations on the mixing ratio α (discrete vs. continuous). However, the parameters of the continuous PGD attack itself (e.g., perturbation budget ε, number of steps, step size) are taken from prior work. Given their importance, a brief sensitivity analysis on these hyperparameters would be valuable. For instance, does a larger ε or more PGD steps in MIXAT yield diminishing returns or even harm utility?
3. The paper demonstrates strong generalization against attacks not seen during training. However, many of these attacks (TAP, PAIR, AutoDAN) fall within the general category of prompt-level modifications. It would be interesting to test MIXAT's robustness against fundamentally different attack modalities, such as those based on role-playing, complex scenarios, or attacks that use non-English languages or encodings, which are not directly related to paraphrasing. This is more of a suggestion for future work but would strengthen the claims of generalizability.

---

> ### Author Rebuttal · Authors · 2025-07-31
>
> We thank Reviewer $\textcolor{orange}{zGMj}$ for the insightful review and are happy to hear that they found our work well motivated and effective, as well as our experimental results convincing. Below, we individually address the reviewer’s questions and comments.
>
> **Q1: Could the authors compare the PAP-MixAT with a variant that uses another discrete attack as seeds for the continuous attacks? Is MIXAT's success tied specifically to PAP or to the general principle of discrete seeding?**
>
> We thank the reviewer for the interesting question. Prompted by the reviewer's response, we additionally train Zephyr and Llama3 MixAT models with statically generated GCG discrete attacks as seeds instead of PAP (using dynamically generated GCG samples would make the training too long and expensive). We present the results below:
>
> | Zephyr Model Utility | ARC-E | ARC-C | MMLU | Harmless | MT-Bench | XSTest |
> |---|---|---|---|---|---|---|
> | CAT | 78.2 | 51.1 | 54.8 | 97.5 | 55.4 | 50.8 |
> | MixAT (PAP + CAT) | 81.4 | 54.0 | 55.8 | 97.5 | 54.3 | 74.0 |
> | MixAT (GCG + CAT) | 81.9 | 54.6 | 56.1 | 90.0 | 56.3 | 49.2 |
>
> | Zephyr Model ASR | Direct R | PAP | TAP | PAIR | AutoDAN | GCG | H Jail | ALO |
> |---|---|---|---|---|---|---|---|---|
> | CAT | 2.5 | 40.0 | 42.5 | 42.5 | 2.5 | 5.0 | 5.0 | 70.0 |
> | MixAT (PAP + CAT) | 0.0 | 0.0 | 0.0 | 0.0 | 0.0 | 12.5 | 5.0 | 15.0 |
> | MixAT (GCG + CAT) | 0.0 | 37.5 | 32.5 | 50.0 | 0.0 | 2.5 | 0.0 | 70.0 |
>
> | Llama3 Model Utility | ARC-E | ARC-C | MMLU | Harmless | MT-Bench | XSTest |
> |----|---|---|---|---|---|---|
> | CAT | 79.7 | 50.9 | 58.0 | 65.0 | 65.7 | 48.4 |
> | MixAT (PAP + CAT) | 80.4 | 50.1 | 59.1 | 85.0 | 55.6 | 40.0 |
> | MixAT (GCG + CAT) | 80.8 | 50.3 | 59.4 | 77.5 | 58.2 | 43.2 |
>
> | Llama3 Model ASR | Direct R | PAP | TAP | PAIR | AutoDAN | GCG | H Jail | ALO |
> |---|---|---|---|---|---|---|---|---|
> | CAT | 0.0 | 30.0 | 47.5 | 70.0 | 0.0 | 7.5 | 5.0 | 82.5 |
> | MixAT (PAP + CAT) | 0.0 | 0.0 | 2.5 | 2.5 | 0.0 | 22.5 | 0.0 | 25.0 |
> | MixAT (GCG + CAT) | 0.0 | 35.0 | 42.5 | 55.0 | 0.0 | 5.0 | 2.5 | 67.5 |
>
> Our results demonstrate that using GCG samples as seeds for the continuous attacks can improve the model robustness over the CAT baseline, but it becomes clear that the lower diversity of the GCG samples is detrimental to the generalization of robustness properties compared to rephrasing attacks such as PAP, TAP, and PAIR. In our MixAT + GCG ablation, we have shown that combining PAP, GCG, and Continuous attacks can further improve robustness, so we hypothesize that the diversity of the attacks used for training is crucial for wide robustness. On the other hand, the comparison between MixAT and our baseline DualAT shows that the stronger mixed (discrete + continuous) attacks are better for training than simply combining the two training losses in a multi-objective manner. We will include this discussion in the next revision of the paper.
>
> **Q2: Could the authors provide some experiments for varying the hyper-parameters of the continuous PGD attack?**
>
> Yes. Below, we show how scaling the $\epsilon$ (0.5x, 2x), the number of steps (0.5x, 2x), and the step-size (0.5x,2.0x) of the continuous PGD attacks used in MixAT affects the robustness and performance of the trained models. Note that for a fair comparison we scaled the step-size along with the perturbation budget $\epsilon$ and we also scaled the step-size inversely with respect to the number of steps.
>
> | Zephyr Model Utility | ARC-E | ARC-C | MMLU | Harmless | MT-Bench | XSTest |
> |---|---|---|---|---|---|---|
> | Mix-AT Original | 81.4 | 54.0 | 55.8 | 97.5 | 54.3 | 74.0 |
> | MixAT 0.5x epsilon 0.5x lr | 82.0 | 55.2 | 56.4 | 97.5 | 53.1 | 63.2 |
> | MixAT 2x epsilon 2x lr | 81.9 | 53.8 | 55.5 | 92.5 | 51.2 | 81.6 |
> | MixAT 2x steps 0.5x lr | 81.8 | 55.1 | 56.0 | 95.0 | 54.5 | 69.6 |
> | MixAT 0.5x steps 2x lr | 81.4 | 54.5 | 56.1 | 97.5 | 52.6 | 56.8 |
>
> | Zephyr Model ASR | Direct R | PAP | TAP | PAIR | AutoDAN | GCG | H Jail | ALO |
> |---|---|---|---|---|---|---|---|---|
> | Mix-AT Original | 0.0 | 0.0 | 0.0 | 0.0 | 0.0 | 12.5 | 5.0 | 15.0 |
> | MixAT 0.5x epsilon 0.5x lr | 2.5 | 0.0 | 2.5 | 2.5 | 0.0 | 12.5 | 2.5 | 12.5 |
> | MixAT 2x epsilon 2x lr | 0.0 | 0.0 | 5.0 | 5.0 | 0.0 | 2.5 | 0.0 | 12.5 |
> | MixAT 2x steps 0.5x lr | 0.0 | 7.5 | 7.5 | 20.0 | 0.0 | 15.0 | 0.0 | 32.5 |
> | MixAT 0.5x steps 2x lr | 0.0 | 2.5 | 10.0 | 10.0 | 0.0 | 10.0 | 2.5 | 20.0 |
>
> Overall, we observe consistent results on the utility benchmarks with only minor and to be expected deviations from the original values. For robustness, this trend continues, however, we find that generally the chosen tradeoff of original MixAT performs on the top end of all tested configurations, with especially high-step low learning rate experiencing a stark drop in ALO robustness. We plan to conduct further hyperparameter experiments here to analyze the trends better, and we will include them in the next revision of the paper.
>
> **Q3: Could the authors test MIXAT's robustness against fundamentally different attack modalities, such as those based on role-playing, complex scenarios, not directly related to paraphrasing?**
>
> Certainly. First we would like to emphasize that rephrasing attacks like TAP and PAP often automatically come up with role-playing scenarios or scenarios where the toxic action request is justified with a particular urgency, necessity, or cultural appropriateness (See Appendix B in [1] for example). Such scenarios are also very common in datasets like HumanJailbreaks, which we also experimented with in Table 1.
>
> Based on the reviewer’s suggestion and to show the generalization of our defense to completely different types of attacks (i.e., neither classical adversarial examples (GCG) nor paraphrasing/role-playing jailbreaks (TAP, PAP, HumanJailbreak)), we conducted two additional experiments - one using code attacks from the QueryAttack dataset [2] on Qwen 2.5 14B models; and one using ASCII art-based attacks from the ArtPrompt dataset [3] on Llama 3 8B models. We chose the model families per attack in such a way that the attacks were particularly potent on the base models to allow us to compare the defense methods meaningfully. We present our results in the tables below:
>
> QueryAttack:
> |model|ASR|
> |---|---|
> |Qwen 2.5 14B Base (HF)|0.965|
> |Qwen 2.5 14B CAT (R)|0.002|
> |Qwen 2.5 14B LAT KL (R)|0.021|
> |Qwen 2.5 14B MixAT|0.000|
> |Qwen 2.5 14B MixAT+GCG|0.000|
>
> ArtPrompt:
> |model|ASR|
> |---|---|
> |Llama 3 8B Base (HF)|0.68|
> |Llama 3 8B CAT (R)|0.02|
> |Llama 3 8B LAT (HF)|0.02|
> |Llama 3 8B MixAT|0.00|
> |Llama 3 8B MixAT+GCG|0.00|
>
> For both experiments, we consider an attack successful if their supplied LLM Judges scored the LLM response with toxicity of 3 out of 5 or higher. For ArtPrompt, we report the results on their best-performing font (vitc-h-gen), while for the QueryAttack, we report the ALO-ASR on the full set of their 9 code modalities. Importantly, we find both MixAT and MixAT+GCG extremely robust to these attacks, not producing even a single toxic answer, unlike CAT and LAT. Further, we see that, in general, the robustness from GCG attacks and role-playing attacks seems to transfer successfully to these settings as well. We will include these results in our revised manuscript.
>
> [1] Mehrotra, et al. "Tree of attacks: Jailbreaking black-box llms automatically.", Neurips 2024
> [2] Zou, et al. "QueryAttack: Jailbreaking Aligned Large Language Models Using Structured Non-natural Query Language.", arXiv 2025
> [3] Jiang, et al. "Artprompt: Ascii art-based jailbreak attacks against aligned llms.",  ACL 2024
>
> **Q4: Could the authors elaborate on the intuition for why the combined perturbation space (R(X) + $\delta$) is so much more effective than the union of the two spaces used in DUALAT?**
>
> The reviewer makes an interesting observation. Intuitively, we found two ways to interpret this phenomenon. First, we note that naturally the space of attacks covered by MixAT is much larger and, unlike the DualAT combination, includes less redundancy (where continuous and discrete give similar signals). Second, and perhaps more importantly, it is natural to assume that especially points around actual (discrete) jailbreak attacks give an informative signal w.r.t. model robustness. That is to say, by applying continuous attacks around a benign data point, we mostly train the model to behave well on inputs where the model already performs well. By shifting this to points that lead to abnormal model behavior, we expect a much stronger signal (in the continuous space around the discrete point) while having to spend less effort optimizing the continuous perturbation. We will include this discussion in the next revision of the paper.
>
> **Q5: Was the time-cost increase of MixAT + GCG training primarily due to the GCG sample generation, or did the training on these combined embeddings also converge more slowly?**
>
> Yes. The increased costs of MixAT + GCG are primarily due to the costs of generating GCG samples during training. We keep the same training schedule (number of steps, number of epochs) for all our MixAT experiments, including MixAT + GCG.
>
> **Q6: Could the authors explain why MixAT's training regimen is less detrimental to model utility compared to CAT?**
>
> Sure. One limitation of CAT is that it relies only on the original set of harmful samples for training, which can be a relatively low-diversity dataset. As a result, the model may start overfitting to superficial patterns in those prompts, such as specific words or phrasing, without fully learning to recognize harmful intent. This often leads to overly cautious behavior, including refusals on benign prompts, as seen in the drop in utility scores and performance on XSTest. In contrast, MixAT includes discrete PAP-style samples during training, which introduces more diverse and realistic adversarial examples, helping the model generalize better and strike a stronger balance between robustness and utility.

---

> > ### Comment · Reviewer_zGMj · 2025-08-05
> >
> > I appreciate the authors' efforts and detailed explanations in the rebuttal. The new experiments resolved my questions. I am keeping my score unchanged, but I am now more confident in this positive assessment.

---

### Decision · Program_Chairs · 2025-09-17

**Decision:**

Accept (poster)

**Comment:**

The paper presents a novel adversarial training method that combines discrete and continuous adversarial training. The idea is impactful, and the results are convincing. I recommend the author to consider the rebuttal and incorporate it in to the paper.